# Soil degradation and herbicide pollution by repeated cassava monoculture within Thailand's conservation region

Ananya Popradit[1]*, Yutthana Nakhokwik[1], Marcel Robischon[2], Shin-Taro Saiki[3], Jin Yoshimura[4,5,6], Arichai Wanasiri[7], Atsushi Ishida[8]*

1 Doctor of Philosophy Program in Environmental Studies, Valaya Alongkorn Rajabhat University Under Royal Patronage, Pathum Thani, Thailand, 2 Department of Agroecology, Faculty of Life Sciences, Albrecht Daniel Thaer Institute for Agricultural and Horticultural Sciences, Humboldt University of Berlin, Berlin, Germany, 3 Forestry and Forest Products Research Institute, Matsuno-sato, Tsukuba, Ibaraki, Japan, 4 Institute of Tropical Medicine, Nagasaki University, Sakamoto, Nagasaki, Nagasaki, Japan, 5 Faculty of Science, Tokyo Metropolitan University, Minami-Osawa, Hachioji, Tokyo, Japan, 6 The University Museum, The University of Tokyo, Hongo, Bunkyo, Tokyo, Japan, 7 Department of National Parks, Wildlife and Plant Conservation, Ban Kho Subdistrict, Non Sang District, Nong Bua Lamphu, Thailand, 8 Center for Ecological Research, Kyoto University, Otsu, Shiga, Japan

* ananya-toon@hotmail.co.th (AP); atto@ecology.kyoto-u.ac.jp (AI)

**Data Availability Statement:** All raw data are within the paper and its Supporting Information files.

## Abstract

In a national park in Northeast Thailand, agricultural land has been converted from natural forest by small-scale farmers for cassava agriculture. We hypothesise that long-termed cassava monoculture leads to the degradation of soil properties. To test the hypothesis, we conducted a five-year (2016–2020) study on the physical and chemical properties of soil in cassava farmland, and also examined the soil properties of its adjacent natural forests, as a control. The examined cassava farmland was converted from the natural forest during the five years from 2011 to 2015. The significant decrease in organic carbon and the increases in exchangeable potassium and bulk density were found in 2016, indicating that these soil properties varied quickly following the farmland conversion. On the other hand, the significant increase in soil nitrogen and the decrease in pH were found later in 2020, indicating that these soil properties were gradually altered by repeated agricultural activities, such as fertilizer application and trampling. In contrast, there were no significant differences in available phosphate, electrical conductivity, cation exchange capacity, and the soil texture (the fractions of sand, silt, and clay) among the forest and farmland soils. The cation exchange capacity was positively correlated to the fraction of clay, the organic carbon, and pH. The use of glyphosate and paraquat herbicides is prohibited within national parks in Thailand. However, in 2020, glyphosate was detected in farmland soil (up to 5.0 mg kg$^{-1}$) during both the rainy and dry seasons, and glyphosate (up to 2.5 mg l$^{-1}$) was detected in stream water from the farmland during the dry season at least in 2020. Soil degradation and herbicide pollution may carry a high risk of causing irreversible changes in terrestrial ecosystems. We discuss the root causes of this issue from perspectives of agricultural production, economy, and the environmental impact, and propose effective policy measures.

**Funding:** This study was supported by funding provided from the Japan Society for the Promotion of Science (No. 16H02708, 23KK0119, 24K03129). The funders had no role in study design, data collection and analysis, decision to publish, or preparation of the manuscript.

**Competing interests:** The authors declare no competing interests.

## Introduction

The global agricultural land area increased by 102 million hectare during the first two decades of the 21st century, corresponding to 9% of the agricultural land area in 2003 [1]. During the 16 years from 2003 to 2019, the per capita agricultural land area decreased by 10% owing to population growth, while the per capita net primary production of agricultural land area increased by 3.5% owing to the enhancement and intensification of agriculture [1]. The increase in crop yields per capita and land area has resulted from the modernization of agriculture. Modernization relies on the development of agricultural facilities and cultivation techniques, development and utilization of agricultural machinery, introduction of improved crop varieties with higher yields, and long-term input of chemical fertilizers and pesticides in farmland. In particular, agricultural modernization is based on large-scale cultivation using a single-crop variety with a high yield (i.e., monoculture).

However, in homogeneous agriculture, diseases and pests are more likely to occur than in traditional agriculture [2]. In such monocultural farmlands, the heavy use of chemically synthesized pesticides and fertilizers is usually required for effective pest control and enhancement of crop yields [3]. In Thailand, the importation of herbicides increased by 10% over a four-year period, rising from 53615 tons in 2009 to 60232 tons in 2012, with glyphosate being the most imported herbicide, both in terms of cost and quantity [4]. Nevertheless, herbicides and pesticides have the potential to cause significant negative impacts on humans (including children), livestock, and the environment if mishandled [5–8]. Continuous attention is thus needed to evaluate and improve regulatory systems and their social implementation. However, in many developing countries, the effectiveness of regulatory systems is often low [6, 9]. Because of its broad effectiveness on both annual and perennial plants (including various crops), glyphosate is popular in Thailand, particularly for effective weed control before planting, even in farmlands [4].

Cassava (*Manihot esculenta* Crantz) is well-suited to tropical and subtropical regions with warm temperatures [10]. This crop plant can grow even in nutrient-poor soil, drought, and heat environments [11] and has significant economic importance for local farmers in tropics [12–15]. Monocultural farming is a common practice worldwide for cassava cultivation. Such monocultural agroecosystems have various risks of soil degradation, including erosion on steep-slope farms, soil compaction and crusting, structural deterioration, flooding, organic carbon loss, microbial decline, salinization, and on-site and off-site damage [16]. Cassava roots penetrate only up to 0.5 m deep into the soil, but can extend 1–2 m horizontally depending on the planting density, contributing to the suppression of soil erosion [17]. However, some reports have showed that the loss of farmland soil is significant especially when cassava roots are harvested [18–20]. On the effects of soil nutrients, cassava extracts significantly less nitrogen and phosphorus from the soil than most other crops while extracting similar amounts of potassium (K) [12]. Consequently, additional fertilization is needed to maintain soil fertility when cassava is continuously cultivated on the same land, especially K [13, 18, 21–23], phosphorus [18, 24, 25], and nitrogen [24], possibly leading to the overuse of fertilizers and an imbalance of nutrient levels within the farmland soil [26]. While once thought to be environmentally friendly, even low-input crops like cassava are now being recognized as contributors to cycles of environmental degradation that endanger future food production [15].

In terms of above-ground environmental factors, including air temperature, solar radiation, and precipitation in Northeast Thailand, the potential production of cassava can be further increased by adopting adequate water management, especially by managing and expanding agricultural land in the mid-highlands [27]. However, we do not have sufficient information regarding soil factors or adequate management based on environmental conservation. In

particular, long-term monitoring of soil quality in cassava farmlands is still rare. Here, we hypothesise that long-term cassava cultivation leads to soil degradation. Additionally, we examined whether herbicides prohibited within Thailand's national parks were used. To sustainably obtain sufficient cassava yields, it is necessary to understand the extent of soil degradation caused by repeated cassava monocultures, and to know the actual impact of the long-term use of chemical fertilizers and pesticides, including herbicides, on soil and water. In the present study, we identify which soil properties incur irreversible damage due to agricultural activities, and furthermore, provide evidence to raise suspicions about the agricultural use of banned glyphosate.

## Materials and methods

### Study site and method overview

Phu Kao-Phu Phan Kham National Park (PKNP) is a national park located on the Khorat Plateau with sandy soil in Northeast Thailand. The study site (16˚55'48.76"N, 102˚27'40.12"E) is located in Phu Kao within this national park. Since 1998, the natural forests within this national park have been designated as protected areas for natural conservation. The Khorat Plateau is primarily composed of nutrient-poor sandy soil. On this plateau, dry deciduous forests develop naturally [28, 29] and are composed of canopy trees that have adapted well to the oligotrophic sandy soils [30]. During the last nine years from 1991 to 2020, the yearly precipitation was 900–1300 mm (1252 mm on average), and the yearly mean air temperature was 21.1–32.5˚C (26.0˚C on average) at approximately 22.7 km southeast of the study site (based on the Thai Meteorological Department). There is a distinct dry season from November to January (Fig 1C).

In Phu Kao, there are three villages (Dong Bak, Wang Mon and Chai Mongkala) surrounded by agricultural lands, which are further enclosed by natural forests. Cassava monoculture has been predominantly carried out on agricultural land, which has been converted from natural forests since the 2000s. In 2010, Thai government established the boundary line between farmland and protected natural forests. At the time, the farmland area (i.e., non-protected area) was approximately 7.3 km$^2$. However, over the successive five years (2011 to 2015), the agricultural area had expanded beyond the boundary line into protected forests, increased by approximately 13 km$^2$. In the present study, during the five years from 2016 to 2020, soil properties were examined at four locations (north, east, south, and west of the main village) in the cassava farmland, which was newly developed the period from 2011 to 2015 (Fig 1A and 1B). To further clarify the agricultural impact on soil, we examined the soil properties of the nearest forests at each of those four locations. Since the examined cassava farmland was converted from the natural forest during the five years from 2011 to 2015, we assumed that the soil in the nearby natural forest retains the original conditions prior to the conversion, thus serving as a control for the farmland soil. The combination of comparisons between farmland and forest soils and long-term monitoring of farmland soil will clarify which soil properties are largely affected by agriculture.

### Soil and water sampling

The farmland soil was collected after crop harvesting in the dry season, when the soil nutrient levels would be lowest in the cassava growing season. Soil samples were collected from at a depth of 15 cm from the soil surface, using a drop-hammer sampler equipped with double cylinders (ANS Inc., American Falls, ID, US). The size of inner cylinder was 4.5 cm in diameter and 15.2 cm in length (approximately 242 cm$^3$ in volume).

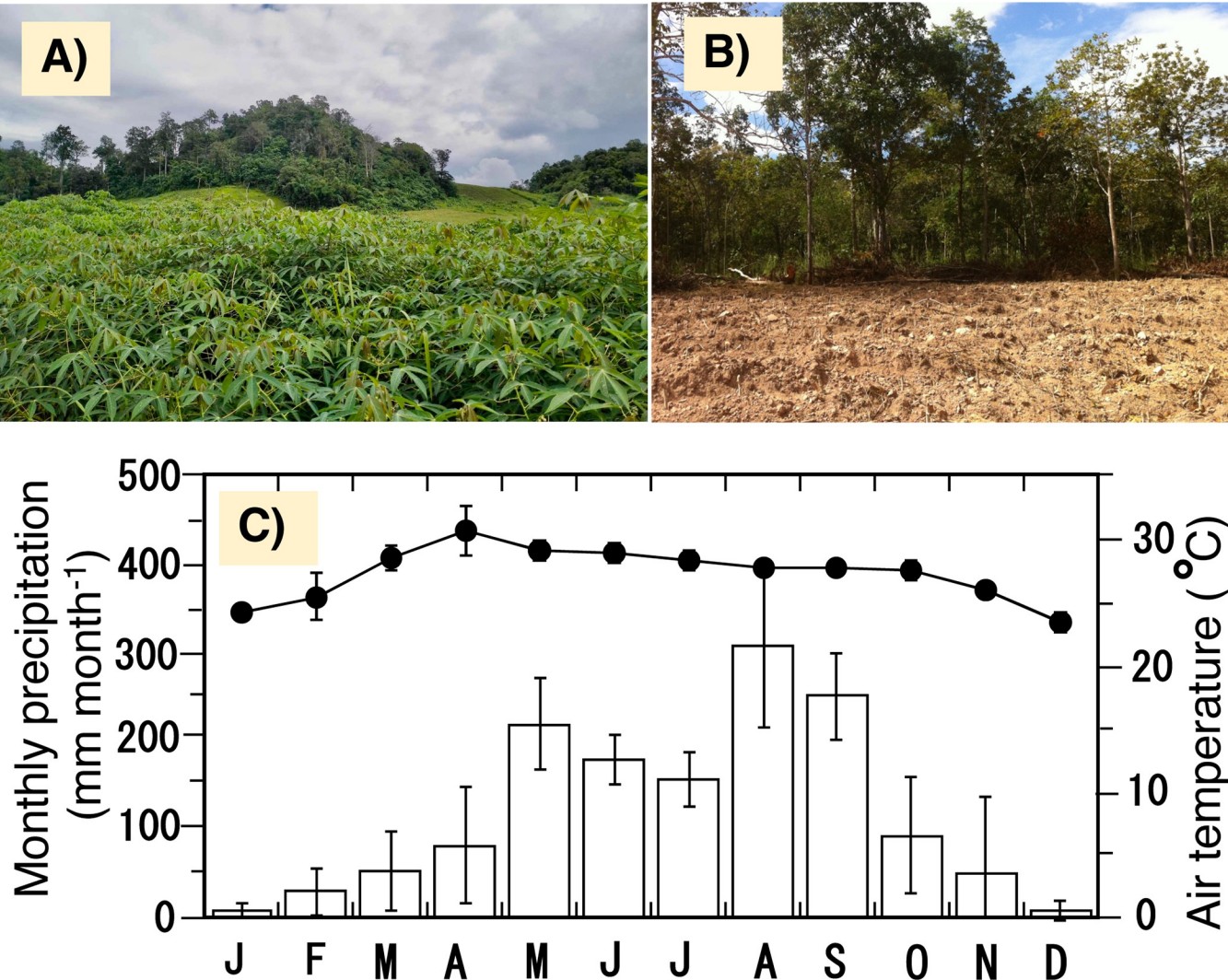

**Fig 1. The photos of monocultural cassava farmland and the seasonal variations of precipitation and air temperature.** The photos are the cassava farmland (**A**) before harvesting in the rainy season and (**B**) after harvesting in the dry season. The forest visible in the background of photos is a protected forest (the photos were taken by A. Popradit, the first author). The research site is the frontline location where the farmland is encroaching into protected forest areas. (**C**) The monthly precipitation (bars) and the monthly averaged air temperature (closed cycles) during the research period (2016 to 2020). Vertical vars show ± 1 S. D.

Soil sampling was systematically conducted at four locations (the north, east, south and west of the main village) with 40 m · 160 m (6400 m²). At each location, the area was further divided into four subplots (40 m · 40 m, 1600 m²); thus 16 subplots were set in total. The positions of the 16 subplots were determined using a Global Positioning System (GPS; Garmin 60Csx, Garmin, Olathe, KS, USA), and the soil (0–15 cm thickness below the ground surface) was collected within each subplot three times (2016, 2018, and 2020). Furthermore, to compare the soil properties between the farmland and forest soils, we investigated the soil properties using soil samples collected from the nearest natural forests at the four locations. The subplots in the forest were also set with the same manner as the farmland, and the soil sampling was conducted one time on November 15 (dry season) in 2020. The sampling number of soil cores was 36 in each subplot, and the soil cores were randomly collected within the subplots.

We examined whether herbicides (glyphosate and paraquat) were used in the farmland. For nature conservation, the use of herbicides within national parks has been prohibited by the Thai government. During the dry season, stream water was blocked in the lower stream to obtain irrigation water. Because of the halted waterflow, pollutant accumulation was predicted during this season. To examine the possibility of herbicide pollution within farmland ecosystems, paraquat and glyphosate were tested in soil and stream water during the rainy (June 6) and dry seasons (November 18) in 2020. The water of the main stream flowing through the agricultural area was collected at three locations (upper, middle, and lower streams) with one sample (1 litter) taken at each location. For the farmland soils, the same samples collected for chemical analysis from the 16 subplots were used.

## Soil analysis

To examine the impact of long-term agricultural land use on farmland soil in Phu Koa, we examined 10 soil properties, soil texture (the fractions of sand, silt, and clay), cation exchange capacity, pH, Kjeldahl nitrogen (N), available phosphorus (P), exchangeable potassium (K), organic carbon, electrical conductivity (EC), and gravimetric soil moisture contents, in monocultural cassava farmland over a five-year period (2016–2020). For soil physical properties, soil texture (dry mass %) and bulk density ($Mg\ m^{-3}$) were examined.

In each subplot, to examine the gravimetric soil moisture contents, bulk density, and the soil texture, we used two soil core samples out of 36 samples for examining each soil property (we show the mean values in each property). In these soil properties, we used the oven-dried soil samples (105°C, 24–48 hours). The soil moisture contents (%) were determining, as follows. The fresh mass of soil within the inner cylinder ($242\ cm^3$) of the drop-hammer sampler was measured using an electronic balance. The soil samples were then dried in an oven and subsequently placed in a desiccator to cool down to room temperature. The gravimetric moisture contents (%) were determined as follows:

$$\text{Soil moisture content} = ((\text{soil wet mass} - \text{soil dry mass})/\text{soil dry mass}) \cdot 100. \qquad (1)$$

The soil bulk density ($Mg\ m^{-3}$) was determining as the ratio of dried soil mass (Mg) to its inner cylinder volume ($m^3$), as follows [31, 32]:

$$\text{Soil bulk density} = \text{dried soil mass}/\text{volume}. \qquad (2)$$

The soil texture (the fractions of sand, silt, and clay) was determining, as follows. The collected soil was sieved with a 2-mm grid mesh, and subsequently oven-dried. The volumetric composition of sand ($> 20\ \mu m$ in diameter), silt ($2–20\ \mu m$) and clay ($< 2\ \mu m$) in the soil samples were determined by the hydrometer method [33, 34] using an ASTM specific gravity hydrometer (Type 152H), based on Stokes' law which expresses the relationship between the settling velocity and particle size [35]. The sedimentation analysis is assumed that the fluid flow can sustain laminar flow, when the particles are spherical and have the same density, and interactions are negligible [36].

To examine the soil's chemical properties, we used air-dried soil from the remaining 30 soil core samples. The soil collected using the soil samplers was thoroughly mixed in the field, and the pooled soil was then divided into four parts. One of these parts was immediately packed into a plastic bag. All collected soil samples were kept in a cool box for transport to our laboratory. In our laboratory, we poured the collected soil into a clean plastic tray and spread it out. While removing other materials such as plant parts, we air-dried the soil during 3–5 days in a room.

The air-dried soil was ground and sifted through a 2-mm grid stainless mesh before chemical analysis. The pH values in the soil solution extracted with distilled water were measured using a pH meter [37]. Soil N concentration (g kg$^{-1}$) was determined using the Kjeldahl method based on a wet combustion [38]. The available P concentration (mg kg$^{-1}$) was extracted with Bray II solution (0.03 N NH$_4$F/0.1 N HCl), and the concentration was colorimetrically examined with a spectrophotometer (U-2000, Hitachi Ltd., Tokyo, Japan) based on the reaction with ammonium molybdate and the development of molybdenum blue colour [39]. The exchangeable K concentration (mg kg$^{-1}$) was extracted using 1 molar NH$_4$OAc solution (pH 7.0), and the extracted K content was colorimetrically examined with a spectrophotometer (XP Flame Photometer, BWB Technologies Ltd, Newbury, UK) [40]. The soil organic carbon (g kg$^{-1}$) was examined using the Walkley-Black chromic acid wet oxidation method, and the oxidizable matter in the soil was oxidized using a 1 N potassium dichromate (K$_2$Cr$_2$O$_7$) solution [41]. Electrical conductivity (EC; mS m$^{-1}$) was measured with an EC meter (Soils Thermal Conductivity Meter, Model MK-TDS210-B, Hangzhou Asmik Sensors Technology Co., Ltd., Zhejiang, China) using a solution extracted with distilled water (1:5). The cation exchange capacity (cmol(+) kg$^{-1}$) was examined using the ammonium saturation method by saturating the soil with a solution of ammonium acetate (NH$_4$OAc, 1 M, pH 7) [42]. NH$_4^+$ replaces the cations that are absorbed by all of the clay micelles in the soil. The amount of NH$_4^+$ was determined by distillation after it was replaced by Na$^+$ from NaCl, which is a direct replacement method.

## Assessing herbicide pollution

The paraquat and glyphosate concentrations in the soil and water were examined with a liquid chromatography-tandem mass spectrometer (UltiMate 3200 QTRAP LC-MS/MS system, AB Sciex Pte. Ltd., Tokyo, Japan). We added 0.5 ml of formic acid to 50 ml of the river water samples and purified through the Osais HLB solid-phase extraction cartridge (Waters Corporation, MA, USA). We added 20 ml of 1% formic acid water/methanol to 20 g of air-dried soil sample (1:1 volume), and the mixture was shaken with a vortex mixer for 5 min. The samples were left to settle for 30 minutes and then centrifuged at 5,000 rpm for 5 min. A 2 ml sample of the supernatant was purified through the Osais HLB solid-phase extraction cartridge. To examine the paraquat and glyphosate concentrations, acetonitrile was used for one-step extraction and dewatering of the samples with MgSO$_4$ and purification with dispersive-solid-phase extraction (dispersive-SPE). Organic acids, other constituents, and excess water were removed using PSA (primary and secondary amine), C18-EC (end-capped) and MgSO$_4$, respectively [43]. These analyses were conducted at 4°C for 24 h.

## Statistics

For each soil property, significant differences between the agricultural and forest soils were examined using the pooled data from each site with one-way analysis of variance (ANOVA) (Table 1). Because the soil samplings were repeatedly conducted at the four locations (north, east, south, west), the statistical differences among four categories (forest, 2016, 2018, 2020) and the four locations were examined using repeated one-way ANOVA (Fig 2 and Table 1). For soil properties with significant differences ($P < 0.05$) in Table 1, multiple comparisons using Tukey's honestly significant difference (HSD) test were furthermore conducted (S3 Table). To evaluate the contribution of the organic carbon, the fraction of clay, and pH to the cation exchange capacity, multiple correlation analysis was conducted, as follows: the cation exchange capacity was the dependent variable, and log$_{10}$(organic carbon), log$_{10}$(clay fraction), and pH were explanatory variables, employing a GLM assuming a Gaussian error distribution with an identity link function (Table 2).

**Table 1. The statistical differences among the sampling years and forest or among the locations for each soil property, using repeated one-way ANOVA.**

| Soil properties | Unit | Expranatory variable | df | Sum Squre | Mean Square | F values | P values | Significance |
|---|---|---|---|---|---|---|---|---|
| Kjeldahl nitrogen (N) | g kg$^{-1}$ | Years and Forest | 2 | 19.99 | 10.00 | 13.93 | **0.000** | *** |
| | | Locations | 3 | 4.10 | 1.37 | 1.90 | 0.144 | n.s |
| Avalable phosphorus (P) | mg kg$^{-1}$ | Years and Forest | 2 | 114.86 | 57.43 | 1.00 | 0.379 | n.s |
| | | Locations | 3 | 149.88 | 49.96 | 0.87 | 0.482 | n.s |
| Exchangeable posassium (K) | mg kg$^{-1}$ | Years and Forest | 2 | 8894.20 | 4447.10 | 1.10 | 0.347 | n.s |
| | | Locations | 3 | 8034.00 | 2678.00 | 0.66 | 0.592 | n.s |
| Organic carbon | g kg$^{-1}$ | Years and Forest | 2 | 78.71 | 39.35 | 24.98 | **0.000** | *** |
| | | Locations | 3 | 26.27 | 8.76 | 5.56 | **0.013** | * |
| pH | | Years and Forest | 2 | 3.15 | 1.58 | 11.97 | **0.000** | *** |
| | | Locations | 3 | 0.45 | 0.15 | 1.15 | 0.370 | n.s |
| Electrical conductivity (EC) | mS m$^{-1}$ | Years and Forest | 2 | 241.29 | 120.65 | 4.04 | **0.025** | * |
| | | Locations | 3 | 257.73 | 85.91 | 2.88 | **0.047** | * |
| Cation exchange capacity | cmol (+) kg$^{-1}$ | Years and Forest | 2 | 1.00 | 0.50 | 1.28 | 0.294 | n.s |
| | | Locations | 3 | 0.51 | 0.17 | 0.44 | 0.730 | n.s |
| Sand fraction | % | Years and Forest | 2 | 52.32 | 26.16 | 3.96 | **0.030** | * |
| | | Locations | 3 | 41.69 | 13.90 | 2.10 | 0.153 | n.s |
| Silt fraction | % | Years and Forest | 2 | 20.76 | 10.38 | 4.98 | **0.014** | * |
| | | Locations | 3 | 9.42 | 3.14 | 1.51 | 0.263 | n.s |
| Clay fraction | % | Years and Forest | 2 | 19.98 | 9.99 | 2.82 | 0.075 | n.s |
| | | Locations | 3 | 11.57 | 3.86 | 1.09 | 0.391 | n.s |
| Bulk density | Mg m$^{-3}$ | Years and Forest | 2 | 0.48 | 0.24 | 46.21 | **0.000** | *** |
| | | Locations | 3 | 0.03 | 0.01 | 1.90 | 0.184 | n.s |
| Moisture content | % | Years and Forest | 2 | 103.03 | 51.52 | 15.93 | **0.000** | *** |
| | | Locations | 3 | 17.92 | 5.97 | 1.85 | 0.192 | n.s |

Significances are shown as follows

***: $P < 0.001$

**: $P < 0.01$, and

*: $P < 0.05$.

All statistical analyses were conducted using R software [44] and RStudio (ver. 2023.06.1; RStudio, Boston, MA, USA). Statistical significance was recognized by $P < 0.05$.

## Results

### Soil properties

All raw data for the farmland and forest soils (0–15 cm depth) are represented in Supplementary Information, S1 and S2 Tables, respectively. The soil properties of the natural forest soil and the annual variations in farmland soil over the five years (2016–2020) are shown in Fig 2 (see Table 1 and S3 Table for statistics). Among the forest and the farmlands soils, no significant yearly variations were found in available P, electronical conductivity, cation exchange capacity, and the soil texture (the fractions of sand, silt, and clay). Compared with the forest soil, the organic carbon in the farmland significantly decreased in 2016, and the soil bulk density significantly increased yearly from 2016 to 2020. In contrast, the significant increase in Kjeldahl N and the significant decrease in pH were found later in 2020. The exchangeable K and soil moisture in the farmland soil trended to increase compared to the forest soils, but the significance ($P < 0.05$) was dependent on the year. Among the locations, no significant

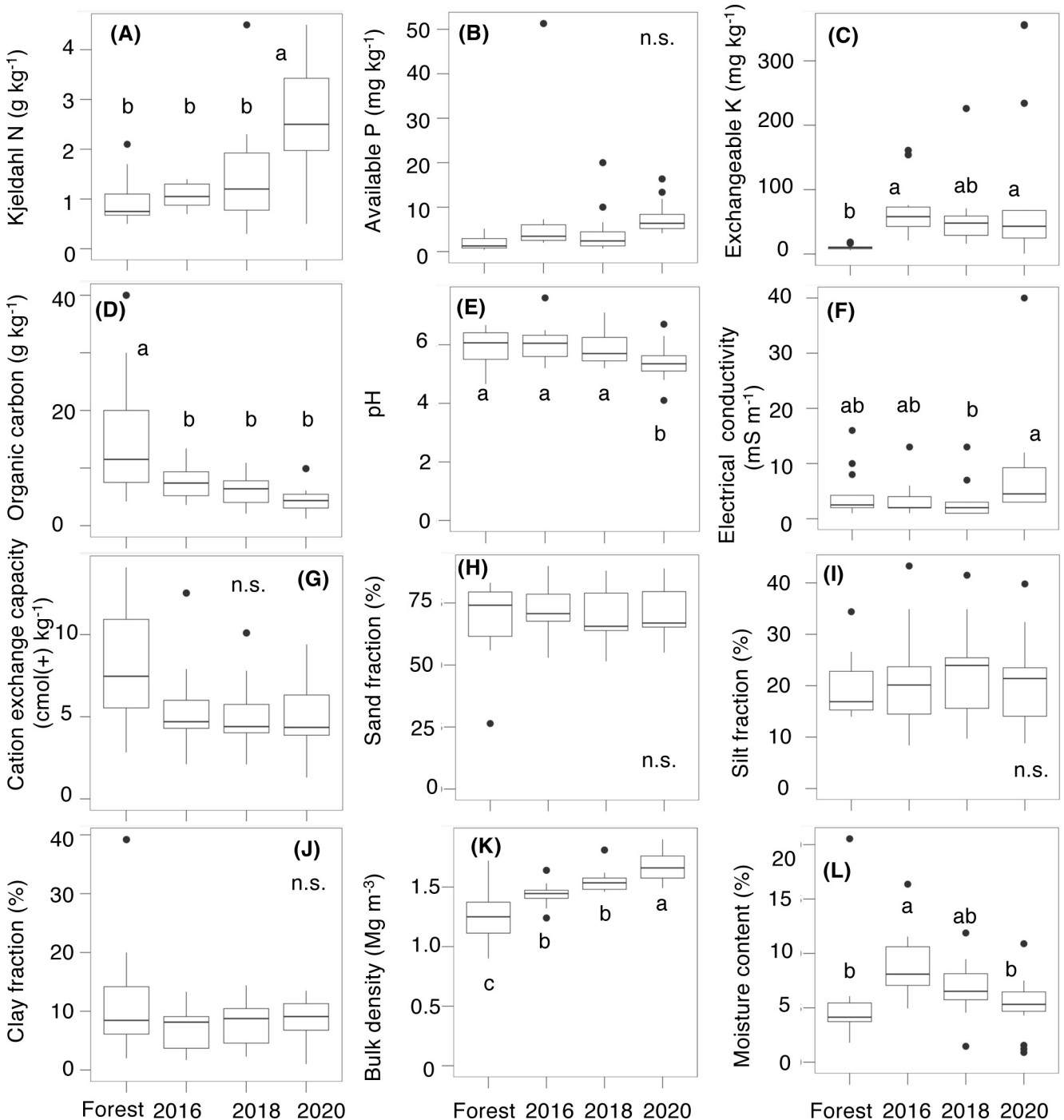

**Fig 2. The properties of the forest soil and the farmland soil in 2016, 2018, and 2020.** (**A**) Kjeldahl nitrogen (the sum of organic N and ammonia N; g kg$^{-1}$), (**B**) available phosphorus (P; mg kg$^{-1}$), (**C**) exchangeable potassium (K; mg kg$^{-1}$), (**D**) organic carbon (g kg$^{-1}$), (**E**) pH, (**F**) electrical conductivity (EC; mS m$^{-1}$), (**G**) cation exchange capacity (cmol(+) kg$^{-1}$), (**H**) fraction of sand (%), (**I**) fraction of silt (%), (**J**) fraction of clay (%), (**K**) bulk density (Mg m$^{-3}$), and (**L**) gravimetric moisture content (%). The boxplot shows the median (horizontal bar), interquartile range (box), 5th and 95th percentiles (whiskers; vertical bars), and outliers (black closed points). Different letters (a, b, c) show the significant differences among the years, as follows: ***: $P < 0.001$, **: $P < 0.01$, *: $P < 0.05$, and n.s.: $P \geq 0.05$.

**Table 2. The results of multiple regression analysis for the cation exchange capacity.**

| Responsible variable | Expranatory variables | Estimated (Standardised partial regression coefficient) | 1 S.D. | T values | P values | Significance |
|---|---|---|---|---|---|---|
| Cation exchange capacity | Intercept | 0.00 | 0.09 | -2.265 | 0.00 | n.s. |
| | $\log_{10}$(organic carbon) | 0.39 | 0.11 | 3.558 | 3.56 | *** |
| | $\log_{10}$(clay fraction) | 0.33 | 0.11 | 3.077 | 3.08 | ** |
| | pH | 0.23 | 0.10 | 2.32 | 2.32 | * |

The cation exchange capacity is the response (dependent) variable, and $\log_{10}$(organic carbon), $\log_{10}$(clay fraction), and pH are explanatory variables. The explanatory variables used are standarised (mean = 0, S.D. = 1). ***: $P < 0.001$, **: $P < 0.01$, *: $P < 0.05$, and n.s.: $P \geq 0.05$.

variations were found for all soil properties, except for the organic contents between the south (S) and north (N) sites (S1 Fig and S3 Table).

Over all, the cation exchange capacity was significantly and positively correlated to $\log_{10}$(organic carbon), $\log_{10}$(clay fraction), and pH (Fig 3). The results of the multiple correlation analysis indicated that the contribution to cation exchange capacity increased in the following order: pH, the fraction of clay, and organic carbon (Table 2).

## Paraquat and glyphosate pollution

In 2020, we assessed glyphosate and paraquat in farmland soil and the water of the main stream in farmland (Table 3). Paraquat was undetectable in both the soil and stream water. Unfortunately, glyphosate in the farmland soil was detected at levels of up to 5.0 mg kg$^{-1}$ in both the rainy and dry seasons throughout at all locations. In the main stream, glyphosate (up to 2.5 mg l$^{-1}$) was observed in the water of the middle stream in the dry season, when waterflow had stopped. In the rainy season, glyphosate was not detected in the stream water, probably because of continuous waterflow. It was confirmed that glyphosate was widely used within farmlands, at least in 2020.

## Discussion

### Soil propeties

The present study clearly demonstrated which soil properties changes rapidly after the farmland conversion (such as organic carbon and bulk density) and which soil properties changes slowly due to subsequent agricultural activity (such as Kjeldahl N and pH). Although the impacts of agricultural managements to soil health have attracted the attention of many scientists, seldom consider the status of forest soil before the agricultural land is established. The cation exchange capacity (CEC) measures a soil's capacity to adsorb and retain cations, showing the degree of soil degradation. In the present study, a decrease trend of CEC was found, but no significant (Fig 2G). Thus, based on CEC, the conspicuous soil degradation was not found during this study period (< 10 years after the farmland conversion). It is known that CEC is affected by clay fraction [45], organic contents, and pH [46, 47]. Since the decreases in the fraction of clay, organic carbon, and pH led to a decrease in CEC in this study (Fig 3), further decreases in organic carbon and pH would result in significant reduction in CEC in the farmland soil.

The importance of soil carbon storage has also attracted the attention of many scientists. For agricultural management, the retention of crop residues and the addition of bioinoculants help to maintain C and N balance in farmland soil, contributing to increasing soil carbon stocks and reducing greenhouse gas emissions [48–51]. Soil bulk density is also an important physical parameter for soil nutrient storage, water transportation and air penetration [48].

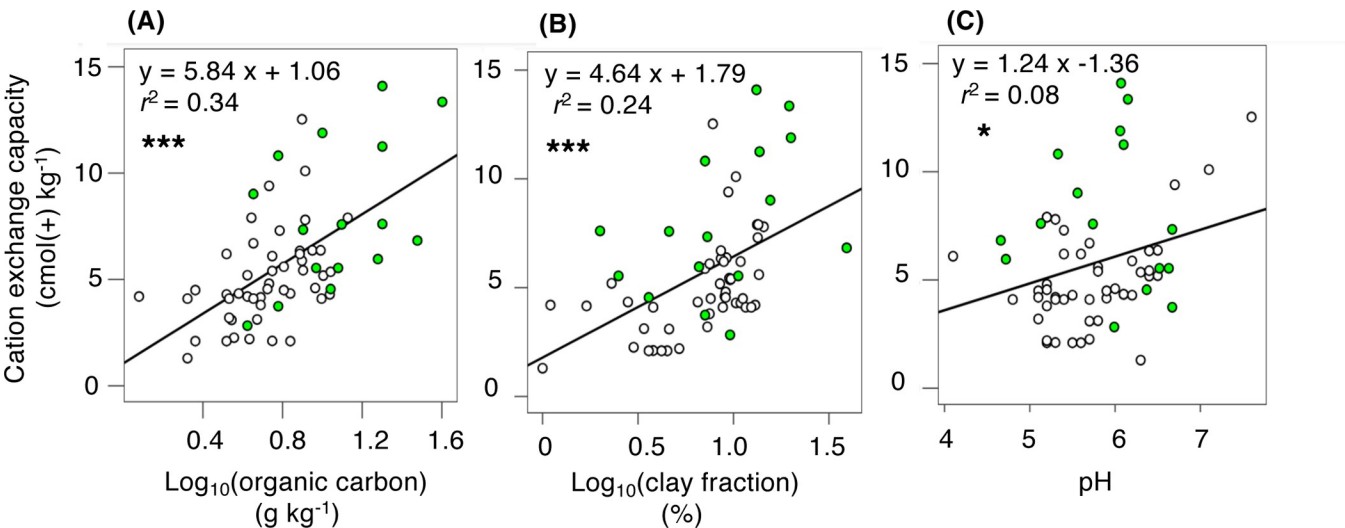

**Fig 3.** Relationships of the cation exchange capacity against (A) $\log_{10}$(organic carbon), (B) $\log_{10}$(the fraction of clay), and (C) pH. Open circles represent data in the farmland soils, and closed green circles represent data in the forest soils. ***: $P < 0.001$, *: $P < 0.05$.

**Table 3. The concentrations of herbicides (paraquat and glyphosate) in the stream water (upper, middle, and lower streams) and the farmland soil.**

|  | Paraquat | | Glyphosate | |
|---|---|---|---|---|
|  | Dry season | Rainny season | Dry season | Rainny season |
| **River water** |  |  |  |  |
| Upper stream | n.d. | n.d. | n.d. | n.d. |
| Middle stream | n.d. | n.d. | **0.5** | n.d. |
| Lower stream | n.d. | n.d. | n.d. | n.d. |
| **Farmland soil** |  |  |  |  |
| North_Subplot 1 | n.d. | n.d. | n.d. | **1.5** |
| North_Subplot 2 | n.d. | n.d. | **1.3** | n.d. |
| North_Subplot 3 | n.d. | n.d. | n.d. | **5.0** |
| North_Subplot 4 | n.d. | n.d. | n.d. | n.d. |
| East_Subplot 5 | n.d. | n.d. | n.d. | **1.5** |
| East_Subplot 6 | n.d. | n.d. | n.d. | n.d. |
| East_Subplot 7 | n.d. | n.d. | n.d. | n.d. |
| East_Subplot 8 | n.d. | n.d. | n.d. | n.d. |
| West_Subplot 9 | n.d. | n.d. | **1.3** | n.d. |
| West_Subplot 10 | n.d. | n.d. | n.d. | **1.5** |
| West_Subplot 11 | n.d. | n.d. | n.d. | n.d. |
| West_Subplot 12 | n.d. | n.d. | n.d. | n.d. |
| South_Subplot 13 | n.d. | n.d. | n.d. | n.d. |
| South_Subplot 14 | n.d. | n.d. | n.d. | **2.5** |
| South_Subplot 15 | n.d. | n.d. | **2.5** | n.d. |
| South_Subplot 16 | n.d. | n.d. | n.d. | n.d. |

The herbicide concentrations were examined in the rainy and dry seasons in 2020. The units of the concentrations in stream water and soil are mg l$^{-1}$ (ppm) and mg kg$^{-1}$ (ppm), respectively. "n.d." means "not detected".

Cassava is often cultivated on tilled plots, traditionally on mounds and ridges with the use of hand hoes or tractor driven implements, significantly increasing bulk density within 0–5 cm layer soil caused by frequent machine movement [52]. The yearly increase in bulk density in this study site (Fig 2K) indicates soil compaction caused by repeated agricultural trampling, possibly reducing cassava growth through inhibition of root penetration and nutrient uptake [53, 54]. Nevertheless, a review paper [52] showed that in a long-term experiment, cassava root yield was increased by non-tillage treatment with mulch residues, with or without fertilizer application. This fact may suggest that it is still possible to develop farming methods that maintain appropriate cassava production while minimizing soil degradation.

### Soil nutrients and acidification

Cassava can grow even in nutrient-poor soil, but to obtain sufficient yields, it is necessary to apply fertilizers containing many elements including K, N, and P [13, 18, 21–25]. In Thailand, non-fertilizer treatment over a period of 15–30 years induces a decrease of 30–40% from the initial cassava yield per unit area [12]. Interestingly, the significant increase in Kjeldal N and the significant decrease in pH were observed with a delayed of 5–10 years after the farmland conversion (Fig 2A and 2E), resulting from that the repeated agricultural activities. The farmers have been continuously applying excess N fertilizer to maintain sufficient cassava yields, compensating for nutrient loss throughout the repeated crop harvesting with artificial fertilization. It is known that additional K fertilization leads to the increases in the number and diameter of cassava roots [23], and soil P and N are the major limiting factors of plant growth, particularly in sandy oligotrophic soil of Thailand [30]. However, the over use of chemical fertilizer leads to soil acidification, which is a major global issue of sustainable development for agricultural ecosystems [55, 56]. The prolonged exposure of cassava roots to high $Al^{3+}$ levels gradually and continuously promotes proton release from their roots and leads to the dissolution of $Al^{3+}$ ions and soil acidification, resulting in high metal toxicity and low P availability [56]. Based on this result [56], the soil pH should be maintained above 5.5 for cassava cultivation. However, pH in the farmland soil was decreasing and the value had dropped to an average of 5.4 in 2020, falling below the critical threshold of 5.5. In long-termed cassava fields (more than 10 years) in the world, the values of pH decreased to 4.3 to 4.4 in Indonesia [57, 58] and to 4.6 in Nigeria [59]. These data indicate that soil pH in cassava farmlands may already be excessively lowered in many regions, especially in repeated cassava monoculture.

Soil degradation advanced by the cassava monoculture resulted in low crop yields, consequently increasing the need for chemical fertilization. Tropical soils generally lack resilience once degraded [60]. Soil degradation and chemical pollution thus pose a high risk of making ecosystems irreversibly unable to return to their original state. Nitrogen loading in the nutrient-poor sandy soil of natural forests in this study site may have unexpected effects on canopy trees that have adapted to oligotrophic soil for a long time [30]. The soil degradation caused by repeated cassava monocultures can thus be a major driver of negative feedback in local-scale environmental conservation. Furthermore, dissolved $NO_3^-$ facilitates leaching from the soil into groundwater and river water.

### Herbicide use

The use of herbicide has increased worldwide. Although the direct toxicity of glyphosate to humans is low and its half-life in soil ranges between 2 and 197 days, the use of glyphosate is still a topic of debate regarding whether it increases the risk of environmental exposure [61]. Furthermore, several glyphosate hotspots in the soil have been detected across the European Union [62]. Since June 30, 1998, the use of all herbicides has been prohibited in all forest

reserves and their watershed areas in Thailand, to prevent hazards and contamination in natural water sources due to spread by wind and water erosion [63–65]. However, the illegal use of glyphosate in this cassava farmland was recognized in 2020. Until now, glyphosate-resistant weeds have been rapidly expanding worldwide [66, 67], and the development of alternative herbicides and low-cost generic herbicides has also been attempted [68, 69]. These herbicides are readily available, and a decrease in pesticide costs may result in an increase in the frequency of its application. Furthermore, because of the quality variations among generic pesticides depending on the manufacturing process and raw materials, it is crucial to ensure their effectiveness and safety through regulation and management on a global scale [9].

## Socioecological perspectives

This case study from Northeast Thailand presented here is a common issue faced by developing countries worldwide generally face. To address this problem and achieve sustainable socio-ecosystems, it is necessary to comprehensively consider the root causes of this issue, including population growth in tropical regions, agricultural production, and the economy, and their impacts on the environment.

Over the last decade, the growth of the human population has resulted in the remarkable expansion of cassava farmland by local-scale farmers, beyond the boundary line between farmland and protected natural forests that set by the Thai government in 2010 [41]. Significant tree species loss was also found within the protected natural forest beside the farmland [28]. The processes of urbanization and agricultural development have had a significant impact on not only the natural environment but also on local socioeconomic factors, including the labour force, education expenses, employment opportunities, household commodities, and vehicles [70–72]. In Thailand, the economic growth rate reached approximately 7.8% per year during the period of the Seventh National Economic and Social Development Plan (1992–1996), whereas the annual growth rate in the country's agricultural sector during the same period was only 2.5% [72]. Certainly, the rate of poverty reduction is closely related to the economic growth rate while facing a consistent increase in income inequality [73]. The proportions of farmland area and forests are 40.7% and 25.3%, respectively, of the country's area (based on the Ministry of Agriculture and Cooperation), and 44.6% of the workforce is involved in the agricultural sector, but contributes only 7.2% of the country's GDP (National Economic and Social Development Board data). Although approximately 67% of the human population resides in rural areas in Thailand (1995–2000), many agricultural lands have been converted into industrial zones, luxury resorts, and towns, resulting in an overall shortage of agricultural land [72]. Nevertheless, a large proportion of the population in Thailand is still engaged in agriculture, which serves as a significant source of income.

To ensure food security and production, the development of new agroecosystems based on environmental conservation is required in the tropics overall. A transition from monocultural cassava cultivation to new farmland use, including the idea of traditional agricultural systems, is necessary. Although traditional slash-and-burn farming has only low productivity, additional chemical fertilizers and herbicides are not needed, having the advantages of low cost and low soil degradation risks [60, 74]. A recent meta-analysis showed that crop diseases decrease and crop yields increase with increasing diversity in intraspecific cultivar mixtures [2]. Another meta-analysis of cassava-based agroecosystems indicated that interspecies cropping had provides positive effects on several key ecosystem services, including pest suppression, disease control, and soil- and water-related services [16]. Experimentally, mixed cropping systems, including intercropping and crop rotation systems, have been shown to play a significant role in increasing cassava yields [17, 75–77]. Thus, diversified agriculture can

thus play a significant role in ensuring a balance between productivity, crop resilience, and environmental health [78, 79]. Furthermore, the use of organic mulch as an alternative to plastic mulch and chemical herbicides is effective for maintaining crop yields through adequate or moderate weed control [80] and soil improvement effects [46], potentially reducing the use of chemical fertilizers and herbicides [81, 82].

Nevertheless, in social implementation, political systems that support income and promote diverse crop cultivation are important, particularly for small-scale farmers. Enhancing ecological agricultural systems, using multiple crops with various adaptive characteristics will be effective not only for maintaining human health and well-being but also for strengthening resilience to climate change, thereby improving food security in developing countries [15, 78]. Adopting agricultural systems that promote ecosystem services enhances resistance not only to pest and disease control but also to climate change, reducing the risk of decreased crop production due to climate change and contributing to the development of a resilient society.

## Conclusion

The comparison of farmland and forest soils, coupled with long-term monitoring of farmland soil, effectively described the change of soil properties resulting from repeated cassava monoculture. The organic carbon rapidly decreased and the bulk density quickly increased due to the farmland conversion. Additionally, a yearly increase in soil bulk density was observed over a 5-year period. In contrast, Kjeldahl N increased and pH decreased with a delayed of 5–10 years after the farmland conversion, due to repeated agricultural activity such as excessive fertilization and repeated trampling. A further decrease in pH values may reduce cation exchange capacity and inhibit cassava growth. Since the farmland examined is located within a national park, the use of herbicides is prohibited. However, we provide evidence of the illicit use of herbicides and the unauthorized expansion of agricultural activities into protected natural forests. For nature conservation, it is necessary to establish political systems that support local farmers financially, promote the transition to diverse crop cultivation, and encourage the use of organic mulch.

## Supporting information

**S1 Table. Raw data for the farmland soil properties.**
(XLSX)

**S2 Table. Raw data for the forest soil properties.**
(XLSX)

**S3 Table. The statistical results of multiple comparisons (Tukey's honestly significant difference test) among the years and the locations.** For only soil properties with significant differences ($P < 0.05$) in Table 1, the multiple comparisons were conducted. ***: $P < 0.001$, **: $P < 0.01$, *: $P < 0.05$, and n.s.: $P \geq 0.05$.
(XLSX)

**S1 Fig. Variations in soil properties among the locations in the farmland.** (A) Kjeldahl nitrogen (the sum of organic N and ammonia N; g kg$^{-1}$), (B) available phosphorus (P; mg kg$^{-1}$), (C) exchangeable potassium (K; mg kg$^{-1}$), (D) organic carbon (g kg$^{-1}$), (E) pH, (F) electrical conductivity (EC; mS m$^{-1}$), (G) cation exchange capacity (cmol(+) kg$^{-1}$), (H) fraction of sand (%), (I) fraction of silt (%), (J) fraction of clay (%), (K) bulk density (Mg m$^{-3}$), and (L) gravimetric moisture content (%). The boxplot shows the median (horizontal bar), interquartile range (box), 5th and 95th percentiles (whiskers; vertical bars), and outliers (black closed points). Different letters (a, b, c) show the significant differences among the years, as follows:

\*\*\*: $P < 0.001$, \*\*: $P < 0.01$, \*: $P < 0.05$, and n.s.: $P \geq 0.05$.
(PDF)

## Acknowledgments

This study was a cooperative research project of the Department of National Parks Wildlife and Plant Conservation(No. 5810701, 5810107, 5921601).

## Author Contributions

**Conceptualization:** Ananya Popradit, Marcel Robischon, Arichai Wanasiri.

**Formal analysis:** Ananya Popradit, Shin-Taro Saiki.

**Investigation:** Ananya Popradit, Yutthana Nakhokwik.

**Project administration:** Ananya Popradit, Arichai Wanasiri.

**Writing – original draft:** Ananya Popradit, Jin Yoshimura, Atsushi Ishida.

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
