## [Decision Letter · Decision Letter 0]

8 Apr 2024

PONE-D-24-04738Soil degradation and herbicide pollution by repeated cassava monoculture within Thailand’s conservation regionPLOS ONE

Dear Dr. Ishida,

Thank you for submitting your manuscript to PLOS ONE. After careful consideration, we feel that it has merit but does not fully meet PLOS ONE’s publication criteria as it currently stands. Therefore, we invite you to submit a revised version of the manuscript that addresses the points raised during the review process.

We look forward to receiving your revised manuscript.

Kind regards,

Khandakar Rafiq Islam, Ph.D.

Academic Editor

PLOS ONE

Journal Requirements:

"This study was supported by funding provided from the Japan Society for the Promotion of Science (No. 16H02708, 23KK0119)."

**Additional Editor Comments:**

Based on reviewers' comments and my evaluation, the authors need to conduct a comprehensive and detailed revision to meet the standards required for further review. Specifically, attention should be given to the following aspects:

The abstract should be focused and clear, providing a succinct summary of the study's objectives, methods, key findings, and implications. The sampling procedure needs to be elaborated upon with greater detail, including the specific name of the sampling method employed. This will enhance transparency and reproducibility. The statistical analyses, particularly those based on ANOVAs, should be thoroughly explained, including the interactions between years and locations. Clear justification for the choice of statistical methods is essential for robust interpretation of results. It is advisable for the authors to seek guidance from a biostatistician to ensure proper organization, analysis, and presentation of the data in tables and figures. This collaboration will enhance the accuracy and validity of the statistical analyses.

Based on appropriate statistical analysis, the results should be presented in tables and figures, accompanied by concise explanations to facilitate understanding and interpretation. Optimization of Figure Scales: Figures should be optimized to ensure clear visualization of the data, with appropriate scaling to accurately represent the findings. A conclusive section summarizing the key findings, implications, and potential avenues for future research should be added to provide closure to the manuscript.

Overall, the manuscript requires significant improvements to meet the standards for publication. By addressing the aforementioned areas and collaborating with a biostatistician, the authors can enhance the scientific rigor, credibility, and relevance of their study, thereby attracting the attention of a wider international audience in the field of soil science.

Reviewers' comments:

Reviewer's Responses to Questions

**Comments to the Author**

1. Is the manuscript technically sound, and do the data support the conclusions?

Reviewer #1: Partly

Reviewer #2: Partly

2. Has the statistical analysis been performed appropriately and rigorously? 

Reviewer #1: Yes

Reviewer #2: Yes

3. Have the authors made all data underlying the findings in their manuscript fully available?

Reviewer #1: Yes

Reviewer #2: Yes

4. Is the manuscript presented in an intelligible fashion and written in standard English?

Reviewer #1: Yes

Reviewer #2: Yes

5. Review Comments to the Author

Reviewer #1: General comments

The study sought to investigate the impacts of cassava monoculture on soil properties and the effects of herbicide application on farmland soils in Thailand. Though the manuscript could be beneficial to the readership of the journal as it elucidates the impact of monoculture on soil fertility variables, it lacks clarity and novelty.

The discussion section of the manuscript is poorly written. The authors largely failed to discuss the results obtained from the study, and this makes the manuscript technically bankrupt to be accepted for publication in its current form.

Below are my specific comments:

Line 89: change “import” to “importation”

Lines 118 – 141: The section should be summarized and moved to the Materials and Methods section of the manuscript.

What is/ are the objective(s) of the study? This should be clearly stated in the introduction section of the manuscript. Also, state the hypothesis/ hypothesis that was/ were tested.

Line 161: Change “Material sampling” to “Soil and water sampling”.

Lines 190 – 191: The import of this sentence is that you equally determined the soil bulk density with the hydrometer method which is not correct. Please rephrase the sentence to make it scientifically correct.

How was the soil bulk density determined?

Lines 197 – 199: This has already been stated in Lines 184 – 186. Please delete.

Line 234: “using”

Lines 256 – 258: Is it part of the statistical analysis? Please delete.

Lines 270 – 271: I am sure you meant to state that the average concentrations of N, P, and K in the farmland soil were 1.9 times, 3.1 times, and 6.3 times higher than those in the forest soil, respectively. Please rephrase the sentence to make it comprehensible.

Lines 300 – 317: This section could be moved to the introduction. You are to discuss the results obtained from the study and not to highlight on general literature.

Lines 321 – 323: Run a simple correlation analysis to substantiate this claim. The cation exchange capacity can equally be affected by the soil pH and organic matter. Authors should correlate the measured soil variables with CEC. Simple correlation coefficients could be used to enrich their discussion.

Line 331 – 334: Rephrase the sentence to make it coherent.

Lines 365 – 442: You did not discuss the results obtained. In a nutshell, the discussion was poorly written. The discussion section needs to be revised.

In conclusion, what are the implications of the findings of the study?

Reviewer #2: The description of weather conditions is not sufficient (lines 147-150). To justify changes in soil conditions, it is necessary to provide a complete description of climatic conditions over the years of research.

In the “Results” section, the authors refer to different seasons (wet and dry), but the years are not specified.

The tables do not indicate the years or period the data was obtained.

The “Conclusions” section is missing. It must be written and inserted into the article.

It is necessary to review the list of references and bring them by the writing rules.

6. PLOS authors have the option to publish the peer review history of their article (what does this mean?). If published, this will include your full peer review and any attached files.

Reviewer #1: No

Reviewer #2: No

---

## [Author Response · Author response to Decision Letter 0]

30 Apr 2024

To Editor,

Thank you for your support for publishing in PLoS ONE. We carefully read the comments from the editor and reviewers, and revised MS for all points. In the attached “Tracking_file”, I show the added or changed parts with yellow-color highlight.

We revised the abstract, and added some information in the Material & Methods. We added two new figures (Figs 2 and 4) for enhancing the visualization of the obtained data. Moreover, we reconsidered and changed the statical approach for data analysis, especially in the multiple comparison (Table 2) and in the correlation analysis related to the cation exchange capacity (Fig 4 and Table 4). 

In the revised MS, we added the hypothesis in the abstract (Line 50-51) and the introduction (Line 126-127). In the discussion part, I newly added discussion related to soil properties (Line 318-370) and the conclusion in the last part of discussion (Line 485-500).

According to the comments, we largely revised the MS. We believe that we are able to enhance the scientific validity and the visualization of the obtained data. We clearly show the key findings in the conclusion. Thank you very much for valuable comments from the editor and reviewers.

Atsushi Ishida, the corresponding author

Reviewers' comments:

Reviewer #1: General comments

The study sought to investigate the impacts of cassava monoculture on soil properties and the effects of herbicide application on farmland soils in Thailand. Though the manuscript could be beneficial to the readership of the journal as it elucidates the impact of monoculture on soil fertility variables, it lacks clarity and novelty.

The discussion section of the manuscript is poorly written. The authors largely failed to discuss the results obtained from the study, and this makes the manuscript technically bankrupt to be accepted for publication in its current form.

Reply: Thank you for your support for publishing in PLoS ONE. We carefully read the comments from the editor and reviewers., and revised MS for all points. In the attached “Tracking_file”, I show the added or changed parts with yellow-color highlight. 

We revised the abstract, and added some information in the Material & Methods. We added two figures (Figs 2 and 4). We reconsidered and changed the statical approach for data analysis. In the Discussion part, I added new discussion related to soil properties and added the conclusion in the last part of discussion.

According to the comments, we largely revised the MS. We believe that we are able to enhance the scientific validity and the visualization of the obtained data. Thank you very much for your valuable comments.

Below are my specific comments:

Line 89: change “import” to “importation”

Reply: I changed it.

Lines 118 – 141: The section should be summarized and moved to the Materials and Methods section of the manuscript.

What is/ are the objective(s) of the study? This should be clearly stated in the introduction section of the manuscript. Also, state the hypothesis/ hypothesis that was/ were tested.

Reply: We moved this part to the head of Material and Methods as a method overview (Line 141-165). In the revised MS, we added the hypothesis in Abstract (Line 50-51) and Introduction (Line 126-127).

Line 161: Change “Material sampling” to “Soil and water sampling”.

Reply: We changed it.

Lines 190 – 191: The import of this sentence is that you equally determined the soil bulk density with the hydrometer method which is not correct. Please rephrase the sentence to make it scientifically correct. How was the soil bulk density determined?

Reply: Sorry, we mistook the statements. We revised the parts in the material and methods (Line 208-211) 

Lines 197 – 199: This has already been stated in Lines 184 – 186. Please delete.

Reply: We deleted it.

Line 234: “using”

Reply: We changed it.

Lines 256 – 258: Is it part of the statistical analysis? Please delete.

Reply: We moved this part from the statistical analysis to the first of the results (Line 270-271).

Lines 270 – 271: I am sure you meant to state that the average concentrations of N, P, and K in the farmland soil were 1.9 times, 3.1 times, and 6.3 times higher than those in the forest soil, respectively. Please rephrase the sentence to make it comprehensible.

Reply: We rephrased it (Line 281-282).

Lines 300 – 317: This section could be moved to the introduction. You are to discuss the results obtained from the study and not to highlight on general literature.

Reply: In the revised MS, we moved this part to the introduction from the discussion (Line 103-111).

Lines 321 – 323: Run a simple correlation analysis to substantiate this claim. The cation exchange capacity can equally be affected by the soil pH and organic matter. Authors should correlate the measured soil variables with CEC. Simple correlation coefficients could be used to enrich their discussion.

Reply: We want to say very thank you for this important suggestion. We added new figure of the correlation analyses for examining the effects of cation exchange on the clay fraction, the organic carbon, and pH (Fig 4). Furthermore, we added a multiple correlation analysis related to Fig 4 (Table 4). According to this analysis and your suggestion, we were able to add significant discussion related to soil degradation (Line 318-370). 

Line 331 – 334: Rephrase the sentence to make it coherent.

Reply: We rephrased it (Line 281-282).

Lines 365 – 442: You did not discuss the results obtained. In a nutshell, the discussion was poorly written. The discussion section needs to be revised.

In conclusion, what are the implications of the findings of the study?

Reply: In the Discussion part, we added new discussion related to soil properties (Line 318-370). In the last part of discussion, we added the conclusion to clarify the key findings (Line 485-500).

Atsushi Ishida, the corresponding author

Reviewers' comments: 

Reviewer #2: 

The description of weather conditions is not sufficient (lines 147-150). To justify changes in soil conditions, it is necessary to provide a complete description of climatic conditions over the years of research.

In the “Results” section, the authors refer to different seasons (wet and dry), but the years are not specified.

The tables do not indicate the years or period the data was obtained.

The “Conclusions” section is missing. It must be written and inserted into the article. It is necessary to review the list of references and bring them by the writing rules.

Reply: Thank you for your support for publishing in PLoS ONE. We carefully read all comments from the editor and reviewers, and revised MS for all points. In the attached “Tracking_file”, we show the added or changed parts with yellow-color highlight. 

We agree with your suggestion that climate data is important. We added climatological data in Fig 1(D). In the annual changes in farmland soil properties, we added the multiple comparisons between the years and the locations, using Tukey’s honestly significant difference (HSD) test (Table 3). In Table 3, more detailed information (including years) is shown.

We added the conclusion in the last part of the discussion to clarify the key findings (Line 485-500).

We adjusted the reference list, according to the rule of PLoS ONE. 

Atsushi Ishida, the corresponding author

---

## [Editor Report · Decision Letter 1]

13 May 2024

PONE-D-24-04738R1Soil degradation and herbicide pollution by repeated cassava monoculture within Thailand’s conservation regionPLOS ONE

Dear Dr. Ishida,

Thank you for submitting your manuscript to PLOS ONE. After careful consideration, we feel that it has merit but does not fully meet PLOS ONE’s publication criteria as it currently stands. Therefore, we invite you to submit a revised version of the manuscript that addresses the points raised during the review process.

While the authors have made some improvements to the paper's quality, they have yet to address the issues previously raised by the editor and reviewers. Without a substantial enhancement in content and quality, further consideration for review in the future cannot be warranted.

The sampling procedure lacks sufficient detail and should include the ***specific name of the employed method*** (such as simple random sampling, systematic sampling, or stratified sampling) to enhance transparency and reproducibility. Additionally, the ***statistical analyses based on sampling method, particularly those utilizing ANOVAs***, need thorough explanation, including the ***interactions between years and locations*** for soil, water and herbicide degradation sampling. Justifying the choice of statistical methods is crucial for robust result interpretation.

Phosphorus should be expressed as "available phosphorus," and potassium as "exchangeable potassium." The method used to measure antecedent soil moisture content, stated as g g^-3^, is unclear and needs clarification. Similarly, the volumetric composition of soil for sand, silt, and clay contents should be elaborated upon, with a reference provided for bulk density determination.

The repeated use of Fig. 2 and Fig. 3 when the data are already presented in Table 3 seems redundant. Combining the data from these figures into one figure, illustrating the variations in soil properties over time for each site (e.g., farmland), would be more informative. The utility of Fig. 4 is questionable without information on correlation coefficients (r) or coefficients of determination (r^2^), which could have been presented in Table 4 as partial regression coefficients. Similarly, the data in Table 4, showing subplot replicated data, could be presented more clearly by categorizing them into east, west, south, and north sites within each of the Upper stream, Middle stream, and Lower stream, respectively or vice-versa.

The Results and Discussion sections require a thorough overhaul. The discussion is notably weak and lacks coherence with the results. Please go thru the attached manuscript file with color marking. There is an overreliance on references, making the paper read more like a review paper. Overall, significant improvements are necessary for the manuscript to meet further review standards.

We look forward to receiving your revised manuscript.

Kind regards,

Khandakar Rafiq Islam, Ph.D.

Academic Editor

PLOS ONE

---

## [Author Response · Author response to Decision Letter 1]

1 Jun 2024

PONE-D-24-04738

Soil degradation and herbicide pollution by repeated cassava monoculture within Thailand’s conservation region

Dear Dr. Ishida,

Thank you for submitting your manuscript to PLOS ONE. After careful consideration, we feel that it has merit but does not fully meet PLOS ONE’s publication criteria as it currently stands. Therefore, we invite you to submit a revised version of the manuscript that addresses the points raised during the review process.

While the authors have made some improvements to the paper's quality, they have yet to address the issues previously raised by the editor and reviewers. Without a substantial enhancement in content and quality, further consideration for review in the future cannot be warranted.

Reply: Thank you for your support for publishing in PLoS ONE. According to your suggestion, we believe that we can enhance the significance of our paper. We revised MS for all pink-highlighted parts, adding more detailed information in the methods. To shortly condense the discussion, we deleted some references and added new discussion related to soil properties, as much as possible. If you could give us advice on further revisions, we would be happy to make those corrections.

The sampling procedure lacks sufficient detail and should include the specific name of the employed method (such as simple random sampling, systematic sampling, or stratified sampling) to enhance transparency and reproducibility. Additionally, the statistical analyses based on sampling method, particularly those utilizing ANOVAs, need thorough explanation, including the interactions between years and locations for soil, water and herbicide degradation sampling. Justifying the choice of statistical methods is crucial for robust result interpretation.

Reply: We set the fixed locations (North, East, West, South), and then we divided into for subplots. And then, we randomly collected soils within the subplot. We added more information in the methods, and added the term of “systematically” (Line 177, Page 8) for the locations and added the term of “randomly” (Line 188, Page 8) within the subplots. For statistics, we did not add the interactive effects between years and locations, because of low sapling numbers. According the comment from the editor, we deleted one figure related to the comparison between the forest and farmland soils (Fig. 1 in the previous version), and combined into one figure in the revised text. In the data of newly combined figure (Fig 2 in the revised MS), we re-calculated ANOVA (Table 2) and then added the multiple comparison test (Table 3). Since the separation among the locations were maintained, we used repeated ANOVA for this statistical analysis. Based on the statistical results of new figure (Fig 2), we largely revised the abstract, results, and discussion in the main text. 

Phosphorus should be expressed as "available phosphorus," and potassium as "exchangeable potassium." The method used to measure antecedent soil moisture content, stated as g g-3, is unclear and needs clarification. Similarly, the volumetric composition of soil for sand, silt, and clay contents should be elaborated upon, with a reference provided for bulk density determination.

Reply: Thank you very much for your suggestion. In the previous MS, there were some mistaken points; sorry. We deleted the term of “volumetric” and changed to the unit of “g g-3” to “%” in the revised MS.

The repeated use of Fig. 2 and Fig. 3 when the data are already presented in Table 3 seems redundant. Combining the data from these figures into one figure, illustrating the variations in soil properties over time for each site (e.g., farmland), would be more informative. The utility of Fig. 4 is questionable without information on correlation coefficients (r) or coefficients of determination (r2), which could have been presented in Table 4 as partial regression coefficients. Similarly, the data in Table 4, showing subplot replicated data, could be presented more clearly by categorizing them into east, west, south, and north sites within each of the Upper stream, Middle stream, and Lower stream, respectively or vice-versa

Reply: Thank you for your suggestion. We deleted the figure related to the comparison between the forest and farmland soils from the main text, and combined into one figure (Fig. 2 in the revised MS). According to the statistical results of the new figure, we partially revised the results and discussion. We added the equations and r2 values in each panel in Fig. 3 and showed the values of partial regression coefficients in Table 4. We revised the form of Table 5, according to your helpful suggestion.

The Results and Discussion sections require a thorough overhaul. The discussion is notably weak and lacks coherence with the results. Please go thru the attached manuscript file with color marking. There is an overreliance on references, making the paper read more like a review paper. Overall, significant improvements are necessary for the manuscript to meet further review standards.

Reply: We revised MS for all pink-highlighted parts, adding some discussion and condensing the paragraphs.

---

## [Decision Letter · Decision Letter 2]

21 Jun 2024

PONE-D-24-04738R2Soil degradation and herbicide pollution by repeated cassava monoculture within Thailand’s conservation regionPLOS ONE

Dear Dr. Ishida,

Thank you for submitting your manuscript to PLOS ONE. After careful consideration, we feel that it has merit but does not fully meet PLOS ONE’s publication criteria as it currently stands. Therefore, we invite you to submit a revised version of the manuscript that addresses the points raised during the review process.

<text x="-9999" y="-9999"></text><path d="M37.5324 16.8707C37.9808 15.5241 38.1363 14.0974 37.9886 12.6859C37.8409 11.2744 37.3934 9.91076 36.676 8.68622C35.6126 6.83404 33.9882 5.3676 32.0373 4.4985C30.0864 3.62941 27.9098 3.40259 25.8215 3.85078C24.8796 2.7893 23.7219 1.94125 22.4257 1.36341C21.1295 0.785575 19.7249 0.491269 18.3058 0.500197C16.1708 0.495044 14.0893 1.16803 12.3614 2.42214C10.6335 3.67624 9.34853 5.44666 8.6917 7.47815C7.30085 7.76286 5.98686 8.3414 4.8377 9.17505C3.68854 10.0087 2.73073 11.0782 2.02839 12.312C0.956464 14.1591 0.498905 16.2988 0.721698 18.4228C0.944492 20.5467 1.83612 22.5449 3.268 24.1293C2.81966 25.4759 2.66413 26.9026 2.81182 28.3141C2.95951 29.7256 3.40701 31.0892 4.12437 32.3138C5.18791 34.1659 6.8123 35.6322 8.76321 36.5013C10.7141 37.3704 12.8907 37.5973 14.9789 37.1492C15.9208 38.2107 17.0786 39.0587 18.3747 39.6366C19.6709 40.2144 21.0755 40.5087 22.4946 40.4998C24.6307 40.5054 26.7133 39.8321 28.4418 38.5772C30.1704 37.3223 31.4556 35.5506 32.1119 33.5179C33.5027 33.2332 34.8167 32.6547 35.9659 31.821C37.115 30.9874 38.0728 29.9178 38.7752 28.684C39.8458 26.8371 40.3023 24.6979 40.0789 22.5748C39.8556 20.4517 38.9639 18.4544 37.5324 16.8707ZM22.4978 37.8849C20.7443 37.8874 19.0459 37.2733 17.6994 36.1501C17.7601 36.117 17.8666 36.0586 17.936 36.0161L25.9004 31.4156C26.1003 31.3019 26.2663 31.137 26.3813 30.9378C26.4964 30.7386 26.5563 30.5124 26.5549 30.2825V19.0542L29.9213 20.998C29.9389 21.0068 29.9541 21.0198 29.9656 21.0359C29.977 21.052 29.9842 21.0707 29.9867 21.0902V30.3889C29.9842 32.375 29.1946 34.2791 27.7909 35.6841C26.3872 37.0892 24.4838 37.8806 22.4978 37.8849ZM6.39227 31.0064C5.51397 29.4888 5.19742 27.7107 5.49804 25.9832C5.55718 26.0187 5.66048 26.0818 5.73461 26.1244L13.699 30.7248C13.8975 30.8408 14.1233 30.902 14.3532 30.902C14.583 30.902 14.8088 30.8408 15.0073 30.7248L24.731 25.1103V28.9979C24.7321 29.0177 24.7283 29.0376 24.7199 29.0556C24.7115 29.0736 24.6988 29.0893 24.6829 29.1012L16.6317 33.7497C14.9096 34.7416 12.8643 35.0097 10.9447 34.4954C9.02506 33.9811 7.38785 32.7263 6.39227 31.0064ZM4.29707 13.6194C5.17156 12.0998 6.55279 10.9364 8.19885 10.3327C8.19885 10.4013 8.19491 10.5228 8.19491 10.6071V19.808C8.19351 20.0378 8.25334 20.2638 8.36823 20.4629C8.48312 20.6619 8.64893 20.8267 8.84863 20.9404L18.5723 26.5542L15.206 28.4979C15.1894 28.5089 15.1703 28.5155 15.1505 28.5173C15.1307 28.5191 15.1107 28.516 15.0924 28.5082L7.04046 23.8557C5.32135 22.8601 4.06716 21.2235 3.55289 19.3046C3.03862 17.3858 3.30624 15.3413 4.29707 13.6194ZM31.955 20.0556L22.2312 14.4411L25.5976 12.4981C25.6142 12.4872 25.6333 12.4805 25.6531 12.4787C25.6729 12.4769 25.6928 12.4801 25.7111 12.4879L33.7631 17.1364C34.9967 17.849 36.0017 18.8982 36.6606 20.1613C37.3194 21.4244 37.6047 22.849 37.4832 24.2684C37.3617 25.6878 36.8382 27.0432 35.9743 28.1759C35.1103 29.3086 33.9415 30.1717 32.6047 30.6641C32.6047 30.5947 32.6047 30.4733 32.6047 30.3889V21.188C32.6066 20.9586 32.5474 20.7328 32.4332 20.5338C32.319 20.3348 32.154 20.1698 31.955 20.0556ZM35.3055 15.0128C35.2464 14.9765 35.1431 14.9142 35.069 14.8717L27.1045 10.2712C26.906 10.1554 26.6803 10.0943 26.4504 10.0943C26.2206 10.0943 25.9948 10.1554 25.7963 10.2712L16.0726 15.8858V11.9982C16.0715 11.9783 16.0753 11.9585 16.0837 11.9405C16.0921 11.9225 16.1048 11.9068 16.1207 11.8949L24.1719 7.25025C25.4053 6.53903 26.8158 6.19376 28.2383 6.25482C29.6608 6.31589 31.0364 6.78077 32.2044 7.59508C33.3723 8.40939 34.2842 9.53945 34.8334 10.8531C35.3826 12.1667 35.5464 13.6095 35.3055 15.0128ZM14.2424 21.9419L10.8752 19.9981C10.8576 19.9893 10.8423 19.9763 10.8309 19.9602C10.8195 19.9441 10.8122 19.9254 10.8098 19.9058V10.6071C10.8107 9.18295 11.2173 7.78848 11.9819 6.58696C12.7466 5.38544 13.8377 4.42659 15.1275 3.82264C16.4173 3.21869 17.8524 2.99464 19.2649 3.1767C20.6775 3.35876 22.0089 3.93941 23.1034 4.85067C23.0427 4.88379 22.937 4.94215 22.8668 4.98473L14.9024 9.58517C14.7025 9.69878 14.5366 9.86356 14.4215 10.0626C14.3065 10.2616 14.2466 10.4877 14.2479 10.7175L14.2424 21.9419ZM16.071 17.9991L20.4018 15.4978L24.7325 17.9975V22.9985L20.4018 25.4983L16.071 22.9985V17.9991Z" fill="currentColor"></path>

Please revise the manuscript according to the reviewer's comments and submit the revised version for publication approval.

We look forward to receiving your revised manuscript.

Kind regards,

Khandakar Rafiq Islam, Ph.D.

Academic Editor

PLOS ONE

Journal Requirements:

Reviewers' comments:

Reviewer's Responses to Questions

**Comments to the Author**

1. If the authors have adequately addressed your comments raised in a previous round of review and you feel that this manuscript is now acceptable for publication, you may indicate that here to bypass the “Comments to the Author” section, enter your conflict of interest statement in the “Confidential to Editor” section, and submit your "Accept" recommendation.

Reviewer #1: All comments have been addressed

Reviewer #2: All comments have been addressed

2. Is the manuscript technically sound, and do the data support the conclusions?

Reviewer #1: Partly

Reviewer #2: Yes

3. Has the statistical analysis been performed appropriately and rigorously? 

Reviewer #1: Yes

Reviewer #2: Yes

4. Have the authors made all data underlying the findings in their manuscript fully available?

Reviewer #1: Yes

Reviewer #2: Yes

5. Is the manuscript presented in an intelligible fashion and written in standard English?

Reviewer #1: Yes

Reviewer #2: Yes

6. Review Comments to the Author

Reviewer #1: General comments

The manuscript has been significantly improved. Though the quality of the writeup is still low, it could be considered for publication. The authors are, however, advised to address the following specific comments:

Lines 50 – 51: Revise the sentence. It could read as “We hypothesise that long-term cassava monoculture leads to the degradation of soil properties”.

Line 111: Replace “harvesting” with “harvested”

Line 126: Long-term

Lines2 215 – 217: The sentence is incomprehensible. Revise it.

Line 308: Delete “not”

Lines 337 and 368: Replace “has” with “is”

Line 369: …leads to…

Lines 379 – 380: “the averaged pH leached to”?

Reviewer #2: All comments sent to the article's authors have been corrected. There are no additional comments.

7. PLOS authors have the option to publish the peer review history of their article (what does this mean?). If published, this will include your full peer review and any attached files.

Reviewer #1: No

Reviewer #2: No

---

## [Author Response · Author response to Decision Letter 2]

15 Jul 2024

Dear Dr. Ishida,

Thank you for submitting your manuscript to PLOS ONE. After careful consideration, we feel that it has merit but does not fully meet PLOS ONE’s publication criteria as it currently stands. Therefore, we invite you to submit a revised version of the manuscript that addresses the points raised during the review process.

Reply: Thank you for your support for publishing in PLoS ONE. The previous MS included some parts of mistyping English. According to the comments of Reviewer 1, we revised all parts. 

Reviewer #1: General comments

The manuscript has been significantly improved. Though the quality of the writeup is still low, it could be considered for publication. The authors are, however, advised to address the following specific comments:

Reply: Thank you for your support for publishing in PLoS ONE. The previous MS included some parts of mistyping English. According to your comments, we revised all parts, as follows： 

Lines 50 – 51: Revise the sentence. It could read as “We hypothesise that long-term cassava monoculture leads to the degradation of soil properties”. 

Reply：　We added the terms of “the degradation of”. (Line 51 in the revised MS)

Line 111: Replace “harvesting” with “harvested”

Reply：　We replace to “harvested”. (Line 111 in the revised MS)

Line 126: Long-term

Reply：　We revised it. (Line 126 in the revised MS)

Lines 215 – 217: The sentence is incomprehensible. Revise it.

Reply：　We revised this part, as follows:” The fresh mass of soil within the inner cylinder (242 cm³) of the drop-hammer sampler was measured using an electronic balance. The soil samples were then dried in an oven and subsequently placed in a desiccator to cool down to room temperature.” (Line 214-217 in the revised MS)

Line 308: Delete “not” 

Reply：　We deleted “not”. (Line 307 in the revised MS)

Lines 337 and 368: Replace “has” with “is”

Reply：　We replaced to “is”. (Line 336 and 367 in the revised MS)

Line 369: …leads to…

Reply：　We added “to”. (Line 368 in the revised MS)

Lines 379 – 380: “the averaged pH leached to”?

Reply：　We revised this part, as follows:” the values of pH decreased to 4.3 to 4.4 in Indonesia [57, 58] and to 4.6 in Nigeria [59].” (Line 378-379 in the revised MS)

Reviewer #2: All comments sent to the article's authors have been corrected. There are no additional comments.

Reply：　Thank you for your support for publishing in PLoS ONE.

---

## [Editor Report · Decision Letter 3]

22 Jul 2024

Soil degradation and herbicide pollution by repeated cassava monoculture within Thailand’s conservation region

PONE-D-24-04738R3

Dear Dr. Ishida,

We’re pleased to inform you that your manuscript has been judged scientifically suitable for publication and will be formally accepted for publication once it meets all outstanding technical requirements.

Kind regards,

Khandakar Rafiq Islam, Ph.D.

Academic Editor

PLOS ONE

Additional Editor Comments (optional):

Please meticulously address the issues cited in the attached reviewed manuscript.
---

## [Editor Report · Acceptance letter]

26 Jul 2024

PONE-D-24-04738R3 

PLOS ONE

Dear Dr. Ishida, 

I'm pleased to inform you that your manuscript has been deemed suitable for publication in PLOS ONE. Congratulations! Your manuscript is now being handed over to our production team.

Kind regards, 

on behalf of

Dr. Khandakar Rafiq Islam 

Academic Editor

PLOS ONE